# Eudraguard^®^ Natural and Protect: New “Food Grade” Matrices for the Delivery of an Extract from *Sorbus domestica* L. Leaves Active on the α-Glucosidase Enzyme

**DOI:** 10.3390/pharmaceutics15010295

**Published:** 2023-01-16

**Authors:** Maria Rosaria Lauro, Patrizia Picerno, Silvia Franceschelli, Michela Pecoraro, Rita Patrizia Aquino, Rosario Pignatello

**Affiliations:** 1Department of Pharmacy, University of Salerno, Via Giovanni Paolo II, 84084 Fisciano, SA, Italy; 2Unesco Chair Salerno, Plantae Medicinales Mediterraneae, University of Salerno, 84084 Fisciano, SA, Italy; 3Department of Drug and Health Sciences, University of Catania, Viale A. Doria, 95125 Catania, CT, Italy; 4CERNUT-Research Centre for Nutraceuticals and Health Products, 95125 Catania, CT, Italy

**Keywords:** Eudraguard^®^, sorbus polar extract, microparticles, spray-drying, dietary supplement, total polyphenol content, DPPH, α-glucosidase inhibition

## Abstract

(1) Background: Eudraguard^®^ Natural (EN) and Protect (EP) are polymers regulated for use in dietary supplements in the European Union and the United States to carry natural products, mask unpleasant smells and tastes, ameliorate product handling, and protect products from moisture, light, and oxidation. Moreover, EN and EP can control the release of encapsulated compounds. The aim of this work was the development, preparation, and control of Eudraguard^®^ spray-drying microparticles to obtain powders with easy handling and a stable dietary supplement containing a polar functional extract (SOE) from *Sorbus domestica* L. leaves. (2) Methods: SOE was characterized using HPLC, NMR, FTIR, DSC, and SEM methods. Furthermore, the SOE’s antioxidant/free radical scavenging activity, α-glucosidase inhibition, MTT assay effect on viability in normal cells, and shelf life were evaluated in both the extract and final formulations. (3) Results: The data suggested that SOE, rich in flavonoids, is a bioactive and safe extract; however, from a technological point of view, it was sticky, difficult to handle, and had low aqueous solubility. Despite the fact that EN and EP may undergo changes with spray-drying, they effectively produced easy-to-handle micro-powders with a controlled release profile. Although EN had a weaker capability to coat SOE than EP, EN acted as a substrate that was able to swell, drawing in water and improving the extract solubility and dissolution/release; however, EP was also able to carry the extract and provide SOE with controlled release. (4) Conclusion: Both Eudraguard^®^ products were capable of carrying SOE and improving its antioxidant and α-glucosidase inhibition activities, as well as the extract stability and handling.

## 1. Introduction

Eudraguard^®^, water-based coating polymers marketed by Evonik Industries AG (EI), meet the regulatory requirements for dietary supplements in the European Union and the United States (GRAS). They can control the oral release profiles of nutraceutical ingredients and improve their bioavailability [1]. In particular, EI reports that Eudraguard^®^ Natural (EN) and Protect (EP) can be used as possible coatings for herbal extracts in solid food supplements. EN is a maize starch-based (starch-acetate) polymer labeled E1420 that is also developed to mask off-flavors and odors [1], and it has gluten-free and certified GMO-free ingredients. EP comprises methacrylic acid and methyl methacrylate monomer units (E1205) (EFSA, 2016). It has also been designed to mask tastes and odors and protect against moisture, light, and oxidation. EP erodes below pH 5.0 and swells at higher pH values, allowing it to dissolve in the acidic stomach environment [2]. There are only a few papers on these coating polymers in the literature. Moreover, such research focuses on the use of the polymers to coat granules and tablets [1] or to encapsulate matrices using different solvent evaporation/emulsion techniques [3,4,5]. None of the papers investigate the microencapsulation ability with spray-drying. Spray-drying technology is widely recognized as easily scalable, with good results in producing stable microsystems in the presence of different coating polymers, such as starches and their derivatives [6,7] or copolymers of Eudragit^®^ [8,9,10]. Furthermore, it is extensively applied to herbal extracts to obtain micro-powders [11,12].

The aim of this work was the physicochemical and technological evaluation of EN and EP microparticles obtained by spray-drying. Furthermore, dietary supplements containing a freeze-dried water extract from *Sorbus domestica* L. leaves (SOE) were developed to evaluate the capacity of both polymers for extract coating and release.

*S. domestica* L. is considered an interesting and rare ornamental, food, and medicinal plant; however, it is an endangered species in many Western countries [13]. The leaves, available during seasonal growth [14], show greater activity for the phenolic fraction than the berries [15]. The leaves are richer in polyphenols, such as flavan-3-class compounds (catechins and proanthocyanidins), flavanols, and phenolic acids, than the fruits [16]. In particular, Matczak et al. [17] identified 44 polyphenols in a polar extract of *S. domestica* L. leaves, demonstrating its antioxidant and anti-inflammatory activity in vitro and plasma oxidative damage protection in vivo. The absence of toxicity against peripheral blood mononuclear cells (PBMCs) also suggests its potential use as a functional ingredient in health products. Furthermore, several studies have shown that the berries and leaves of *S. domestica* have diuretic, anti-inflammatory, antidiabetic, anti-atherogenic, and antidiarrheal activities [18]. However, despite the therapeutic potential of the leaves, only the fruits are commonly used.

To confirm the leaves’ health safety and potential health benefits, an aqueous extract of *S. domestica* L. leaves (SOE) was prepared and characterized using chromatographic, spectrophotometric, and spectroscopic techniques. In vitro antioxidant/radical scavenging activity (DPPH test), α-glycosidase inhibition, and cell viability (MTT assay) were also examined. It was due to the potential use of SOE as functional ingredient in the design of human health products that a cell viability test (MTT assay) was also carried out [19].

Finally, to evaluate the polymers’ ability to release and protect SOE, all the raw materials and formulations developed were characterized with UV, HPLC, DSC, FTIR, and SEM techniques. Water dissolution tests and accelerated stability studies were also performed to evaluate the shelf life of the final products.

## 2. Materials and Methods

### 2.1. Materials

The leaves of *S. domestica* L. were collected in the Salerno area of Italy in September 2020 and identified by one of the authors. A voucher sample (SD01) was deposited at the Herbarium of the Department of Pharmacy, University of Salerno. Samples of EN and EP were supplied by Evonik Industries (Evonik Rohm GmbH, Darmstadt, Germany). Unless otherwise specified, all reagents and compounds, as well as the α-glucosidase enzyme (Saccharomyces cerevisiae type I, lyophilized powder, ≥10 units/mg protein), were purchased from Merck Life (Milan, Italy). All the other chemicals used in the study were analytical grade. Immortalized human normal keratinocyte (HaCaT) cell lines and all the supplements for cell cultures were obtained from Gibco Life Technology Corp. (Thermo Fischer Scientific, Milan, Italy).

### 2.2. General Experimental Procedures

NMR spectra were determined with a Bruker DRX-600 NMR spectrometer in deuterated methanol (CD_3_OD). Chemical shifts are expressed in parts per million (δ) using tetramethylsilane (TMS) as an internal standard [19]. Electrospray ionization mass spectrometry (ESIMS) was performed using a Finnigan LCQ Advantage instrument (Thermo Scientific, San Jose, CA, USA). Column chromatography was performed using a Sephadex LH-20 (Pharmacia, Uppsala, Sweden). Thin-layer chromatography (TLC) was performed on precoated silica gel 60 F254 plates (Merck Life, Milan, Italy), and the spray reagent cerium sulfate and UV (254 and 366 nm) were used for the spot visualization. Semi-preparative HPLC separation was performed with a Waters 590 pumping system equipped with a Waters R401 refractive index detector, a Rheodyne injector (100 μL loop; Waters, Corp., Milford, MA, USA), and a Phenomenex Luna C_8_ column (250 × 10 mm i.d, particle size 10 μm; Phenomenex Inc., Castel Maggiore (BO), Italy). Quantitative HPLC analysis was carried out with an Agilent 1260 system (Agilent Technologies, Waldbronn, Germany) equipped with a Model G-1312B pump, a Model G-4225A degasser, a Rheodyne Model G-1322A loop (20 μL), and a DAD G-4212A detector using a Nucleodur 100-5 C_18_ EC column (150 × 4.6 mm, 5 μm, Delchimica, Naples, Italy). Peak areas were calculated with an Agilent integrator (Agilent Technologies, Waldbronn, Germany).

### 2.3. Methods

#### 2.3.1. Pre-Formulation Studies

To develop oral formulations, evaluation of the physicochemical (SOE yield extraction process, qualitative and quantitative analyses) and technological (calculation of SOE marker equivalents, SOE water solubility and in vitro dissolution rate, raw materials’ morphology) characteristics of all the raw materials was carried out.

##### Physicochemical Characterization

Extraction and yield of the process (SOEY). Air-dried leaves of *S. domestica* (250.0 g), powdered and defatted with n-hexane and chloroform, were extracted at room temperature with methanol (3 × 800 mL for 24 h). The organic solvent was removed with a rotary evaporator (Rotavapor Hei-Vap Core, Delchimica, Naples, Italy) under a vacuum at 40 °C. Then, the extract was suspended in water and lyophilized (Alpha 1-2 D freeze dryer Martin Christ, Osterode am Harz, Germany) to obtain the solid SOE residue (39.3 g). The lyophilized material yield (SOEY) was determined with a gravimetric method (CAL-Gibertini (max 110 g, d = 0.1 mg; +15 °C/30 °C).

Qualitative analyses. An aliquot (2.5 g) of SOE was purified with a Sephadex LH-20 column (1 m × 3 cm, Pharmacia) using MeOH as the eluent (flow rate 0.8 mL/min). Fractions (8 mL each) were collected, analyzed with TLC ((Si-gel, n-BuOH–AcOH–H_2_O (60:15:25), CHCl_3_–MeOH–H_2_O (7:3:0.3)), combined into four majority fractions (I–IV) based on TLC pattern, and purified with RP-HPLC (mobile phase, MeOH–H2O, 50:50 *w*/*w*).

Quantitative analyses and total phenolic content (TPC). A quantitative HPLC analysis was conducted using water (solvent A) and MeOH (solvent B), both containing 0.1% (*v*/*v*) formic acid, as an eluent system. The elution gradient used was as follows: 0→10 min, 35→37% B; 10→15 min, 37→50% B. Analysis was carried out in triplicate at a flow rate of 0.8 mL/min with a DAD detector set at 366 nm. Three different rutin reference standard solutions [5] were prepared in the 0.25–1.00 mg/mL range and analyzed using the linear least-squares regression equation derived from the peak area (regression equation: y = 22,499x − 579.5, r^2^ = 0.999, where y is the peak area and x the concentration). The peaks associated with compounds **3** and **5** were identified by their retention times and UV spectra and confirmed with co-injections. SOE was dissolved in MeOH and analyzed under the same chromatographic conditions. Total polyphenolic content (TPC) was determined according to the Folin–Ciocalteu colorimetric method [19] and expressed as the gallic acid equivalent (GAE mg/g of dry extract).

##### Raw Materials’ Technological Characteristics

SOE marker calculation. All the analyses were carried out using a UV-Vis online apparatus (UV–Vis 1601 Shimadzu Europa, Duisburg, Germany; maximum of 366 nm, 1 cm cell). Quercetin equivalents (Q), chosen as the UV marker, and SOE concentration were calculated according to ICH (Q2 (R1)) guidelines (percent relative standard deviation statistical validation (% RSD)) using the Lambert–Beer law (USP 37) as follows:E1%= 1 cm × c × l(1)
where E1% 1 cm is the absorbance of 1 g/100 mL (1% *w*/*v*) solution in a 1 cm cell; c is the concentration of the solution (g/100 mL); and l is the cell pathlength of the held sample.

*SOE water solubility:* Excess extract (120.0 mg) was introduced into a flask containing 12 mL of water. The sample was shaken for 3 days at 25 °C (the shake flask method) [20] and filtered (0.45 μm filter), and the supernatant solution was examined at 366 nm (UV–Vis apparatus, 1 cm cell) to determine the amount of dissolved extract. Each analysis was carried out in triplicate.

*SOE* in vitro *dissolution test:* In this test, 3.0 g of SOE, corresponding to 300.0 mg of Q, was dissolved in 1000 mL of water (SOTAX AT Smart Apparatus, Basel, Switzerland) and assessed with an online spectrophotometer at 366 (Lambda 25 UV–Vis spectrometer, Perkin-Elmer Instruments, MA, USA) and USP 37 dissolution test apparatus (n.2: paddle, 100 rpm at 37 °C). All the dissolution/release tests were carried out in triplicate under “sink conditions”. In the graph, the mean values are reported (standard deviations < 5%).

*SOE and pure polymers’ morphology:* A scanning electron microscope (SEM) (Carl Zeiss EVO MA 10, Carl Zeiss s.p.a., Milan, Italy), operating at 20 kV, was used to analyze the raw materials. Dried samples were dispersed on adhesive carbon tabs (12 mm) coated with aluminum stub and metalized with a LEICA EMSCD005 sputter coater (sputter current: 30 mA; sputter time: 135 s; thick gold layer: 200–400 Å).

*SOE cell viability assay*. HaCaT cell lines were grown in Dulbecco’s modified Eagle medium (DMEM) containing high glucose with 10% FBS. Cell viability was evaluated by means of MTT ([3-(4,5-dimethylthiazol-2-yl)-2,5-diphenyltetrazolium bromide]) to compare the effects of potentially cytotoxic substances with a control condition [21]. Approximately 5.0 × 10^4^ cells per well were plated on 96-well plates and allowed to adhere for 24 h. Then, the medium was replaced with fresh medium alone or medium containing different concentrations of SOE (12–24 mg/mL), and cells were incubated for 48 h. Finally, the plates were centrifuged at 1200 rpm for 5 min, the medium was aspirated, and 100 μL of 1 mg/mL MTT was added to each well. The plates were kept at 37 °C for the time necessary for the formation of formazan salts (1–3 h depending on cell type). The solution was then removed from each well, and the formazan crystals within the cells were dissolved with 100 μL of DMSO [22]. Each well’s optical density (OD) was measured with a microplate spectrophotometer (Multiskan Spectrum Thermo Electron Corporation reader) equipped with a 620 nm filter. The viability of the cell lines in response to treatment-tested samples was calculated as follows:% viability = (OD treated/OD control)] × 100

#### 2.3.2. Formulation Studies: Development and Preparation of Microparticles

*Feed preparation:* Due to the basic nature of EP, an acidic aqueous solvent was chosen to obtain a clear to slightly cloudy solution [23]. EN, a starch derivative, was suspended in hot water (80 °C) to obtain a fine dispersion.

Three samples of each aqueous polymeric feed (1% m/V) were prepared in triplicate with stirring for 24 h as follows:(1)2.0 g of EN or EP in 200 mL of deionized water or acid water (pH 1.0), respectively;(2)2.0 g of EN or EP and 0.5% glycerol in 200 mL of deionized water or acid water (pH 1.0), respectively;(3)2.0 g of EN or EP and 1% Tween 60 in 200 mL of deionized water or acid water (pH 1.0), respectively.

Then, SOE was added to sample 1 with a 1:1 extract/polymer ratio and stirred for 10 min. All the feeds were spray-dried (Mini Buchi B-290, Flawil, Switzerland).

*Spray-drying parameters:* Nozzle: 700 µm; inlet T: 120 °C; outlet T: 65 °C; pump: 10; flow rate: 3 mL/min; aspirator: 100.

#### 2.3.3. Physicochemical and Technological Characterization of Microparticles

*Spray drying process yield (SPY), actual active content (AAC), and inclusion efficiency (IE).* First, 5.0 mg of SOE or microparticles was dissolved directly in 5 mL of deionized water and vortexed for 60 s at 3000 rpm to determine the actual active content (AAC) using the previously reported HPLC method (see “Physicochemical Characterization” paragraph). AAC was expressed as the quercitrin (Qcn) and rutin (Rt) content percentage equivalent in 100 mg of powder. Each analysis was carried out in triplicate, and the results were expressed as mean values.

The IE was calculated from the ratio of actual extract content (AEC) to theoretical extract content (TEC) in a freeze-dried complex according to the following equation:IE (%) = (AEC/TEC) × 100

*Fourier-transform infrared spectroscopy (FTIR*). FTIR spectra were analyzed from 2000 to 600 cm^−1^ with 256 scans and 1 cm resolution (IRAffinity-1S, Shimadzu Corporation, Kyoto, Japan, MIRacle ATR with ZnSe thin crystal).

*Differential scanning calorimetry (DSC).* The thermal behavior of each sample was analyzed using DSC with an indium-calibrated Mettler Toledo DSC 822e (Columbus, OH, USA) with one thermal cycle. The samples were placed in a pierced 40 μL aluminum pan and scanned (10 C/min) between 25 and 350 °C. The melting temperature (Tm) and heat of fusion (DHm) were measured.

*Microparticles’ morphology.* As for the SOE, morphology was investigated using scanning electron microscopy (see “Raw Materials’ Technological Characteristics” paragraph). The particle diameters were determined from an average of at least 20 observations.

*Solubility study (SS) and dissolution rate (DR).* SS and DR analyses were conducted in triplicate (standard deviation <5%) as previously reported for SOE (see “Raw Materials’ Technological Characteristics” paragraph) using amounts of microparticles corresponding to 0.3 mg of Q.

#### 2.3.4. In vitro Biological Activity

*DPPH test.* The free radical scavenging activity of the SOE and compounds was assayed using the 1,1-diphenyl-2-picrylhydrazyl radical (DPPH) method following [19]. The DPPH concentration in the reaction medium was calculated with linear regression analysis in GraphPad Prism 7 software (San Diego, CA, USA), and the mean effective scavenging concentration (EC_50_) was determined as the concentration of sample necessary to decrease the initial DPPH concentration by 50%. A higher free radical scavenging activity was indicated by a low EC_50_ value.

*α-glucosidase test.* The antidiabetic activities of SOE and its constituents were evaluated with an α-glucosidase assay following [24]. A mixture containing 10 μL of the sample at different concentrations, 320 μL of potassium phosphate buffer solution (PPBS, 0.1 M, pH 6.8), and 50 μL of the substrate (5 mM 4-nitrophenyl α-D-glucopyranoside in 0.1 M PPBS) was incubated at 30 °C for 15 min. Then, 20 μL of enzyme (Saccharomyces cerevisiae) solution (0.5 U/mL in 0.1 M PPBS) was added to start the reaction. After incubation, the reaction was stopped by adding 3.0 mL of 0.05 M sodium hydroxide. The absorbance was measured at 410 nm. A negative control (vehicle in place of the sample) and blank (PPBS in place of the enzyme for correction of the background absorbance) were prepared using the same procedure. The concentration of the sample that inhibited the enzyme’s activity by 50.0% (IC_50_) was calculated with nonlinear regression analysis in GraphPad Prism 7 software (San Diego, CA, USA).

#### 2.3.5. Accelerated Stability Test According to ICH Guidelines and Functional Activity of ENSOE and EPSOE

The following conditions for the stability test were adopted: 40 °C ± 2 °C/60%RH ± 5%RH (ICH, 2003, Climatic and Thermostatic Chamber, Mod.CCP37, AMT srl, Milan, Italy). The samples were analyzed using HPLC, FTIR, and DSC according to the conditions described for the methods (Section 2.3.1 and Section 2.3.3). Moreover, the α-glucosidase inhibition of the samples dissolved in distilled water (from 20.0 to 3.0 μg/mL) was evaluated according to the experimental conditions described in Section 2.3.4. Finally, the free-radical effect of the samples (from 100.0 to 50.0 μg/mL) dissolved in a mixture of MeOH:H_2_O (1:1, *v*/*v*) was assayed using the DPPH• radical (Section 2.3.4).

#### 2.3.6. Statistical Analysis

In this study, all experimental data are reported as the means ± standard deviation (SD) from three independent experiments performed in triplicate. Statistical analysis was undertaken using ANOVA followed by the Bonferroni test. All calculations were performed using GraphPad Prism 7 software for Windows (San Diego, CA, USA). A *p*-value < 0.05 was considered statistically significant.

## 3. Results and Discussion

### 3.1. Pre-Formulation Studies

#### 3.1.1. SOE Extraction and Characterization

A freeze-dried polar extract from *S. domestica* leaves (SOE) was produced (yield: 15.7 ± 1.8% *w*/*w*). To guarantee the reproducibility of the extraction methods and biological activity, chemical standardization of the dry extract with potential use as an active dietary supplement ingredient was needed. Therefore, four fractions and six major compounds (Appendix A) were isolated from the SOE extract through molecular exclusion chromatography and RP-HPLC.

Fraction I (160.0 mg) yielded compounds 1 (2.0 mg, tR = 16 min) and 2 (2.3 mg, tR = 19 min). Fraction II (50.3 mg) yielded compounds 3 (15.0 mg, tR = 13.0 min), 2 (6 mg), and 4 (7.3 mg, tR = 28.0 min). Fraction III (43.9 mg) yielded compound 5 (5.7 mg, tR = 24 min), while fraction IV (200.1 mg) yielded compounds 1 (3.1 mg, tR = 16.0 min), 6 (12.0 mg, tR = 18.0 min), and 5 (18.6 mg).

Their structures were elucidated by comparing nuclear magnetic resonance (NMR) and electrospray ionization mass spectrometry (ESIMS) data to data found in the literature, as isoquercitrin (1) [25], quercetin 3-(2″-glucosyl)-rhamnoside (2) [26], rutin (3) [27], quercetin 3-(2″-xylosyl)-rhamnoside (4) [28], quercitrin (5) [25], and hyperoside (6) [29].

All isolated compounds were reported as characteristic polyphenols occurring in the *Sorbus* genus [17] that play pivotal roles in the extracts’ activity [30].

#### 3.1.2. SOE Technological and Biological Characterization

SOE was rich in quercetin (Q) derivatives, such as rutin (Rt) and quercitrin (Qcn). For this reason, Q was chosen as the UV marker, while Qcn and Rt were chosen as HPLC markers. The UV–Vis analysis showed an amount of Q of 10% in the SOE, while the Qcn and Rt contents were 2.4% ± 0.1 and 2.9% ± 0.2, respectively, according to the HPLC analysis (Appendix A, Table 1).

Organoleptic examination of SOE and its various aspects revealed an extract that was challenging to handle, slightly soluble, sticky, and with a pungent odor. The SOE’s morphology, water solubility, and in vitro dissolution rate were tested because of the influence of in vivo absorption and bioavailability. The SOE raw material (Figure 1) looked like a large (70.0 μm), crystalline, shiny mass with a discontinuous outline.

The solubility test, carried out according to USP 37, showed that SOE was weakly soluble in water (4.94 g/L ± 0.6 mg/L) at room temperature; the water dissolution test demonstrated that no more than 24.0% of the SOE could be dissolved in 45 min. These properties may affect the in vivo behavior, reducing the extract’s in vivo absorption and bioavailability.

Previously, the absence of cytotoxicity for the polar extracts of the leaves of *S. domestica* in a model of human peripheral blood mononuclear cells (PBMCs) has been reported [17]. To confirm this finding, SOE was tested with the in vitro MTT assay, an indicator of cell viability, against normal cell lines (HaCaT). The results showed that cell growth was not reduced after 48 h of incubation with different concentrations (12–24 mg/mL) of SOE (Appendix A), evidencing the extract’s safety.

#### 3.1.3. Polymer Technological Characterization

With the aim of improving the extract handling and technological characteristics and obtaining SOE dietary supplements, we considered EP, a basic methacrylate copolymer (BMC), and EN, an acetylated starch derivative, as food grade matrices for the oral delivery of SOE.

The food grade of the above polymers was established on the basis of the following studies. The European Food Safety Authority considers that basic methacrylate copolymer (BMC) has a negligible mass below 1000 D [31]. One study found that less than 0.02% of the administered dose of EP was absorbed. No absorption by the body—and, thus, no degradation—was observed [23]. Therefore, the metabolism/toxicokinetic studies did not suggest any toxic behavior [23,31]. Furthermore, the EFSA established the margin of safety (MOS) ranges for heavy users as 43 to 85 for adults and 63 to 125 for children. The FDA estimated an acceptable daily intake (ADI) for BMC as a food additive of up to 11.7 mg/kg bw in adults and 13.3 mg/kg bw in children. For EN, long-term carcinogenicity studies showed no significant effects in terms of histo-pathological changes in rats or humans after 60 g (860 mg/kg bw) of acetylated starch was consumed over four successive days [32].

The morphological analysis of EN and EP using SEM showed that pure EN (Figure 1) was made up of thin and crystalline structures of about 500 μm with irregular shapes and jagged edges, while pure EP (Figure 1) showed smaller crystalline structures (10–50 μm) and a wrinkled surface with the same edges as EN.

### 3.2. Formulation Studies and Characteristics of Spray-Dried Microsystems

#### 3.2.1. Characteristics of Spray-Dried SOE (SOE sd) and Blank Microparticles (EN sd and EP sd) with Respect to Raw Materials

The SOE sd yield was only 3%, with an AAC of 80% for both Rt and Qcn, of the yield for raw SOE. The loss of mass suggests that, during spray-drying processing, the extract adhered to the dry chamber apparatus, probably due to the presence of sugars. Despite the improvement to the solid state of the raw SOE resulting from the spray-drying process, which made it more amorphous, the extract still remained sticky post-spray-drying process (SOE sd, Figure 2).

For the blank EN and EP production, the yields were 65.0 ± 1.5 and 71.0 ± 2.0, respectively. The microscopy observations of the EN sd and EP sd microsystems (Figure 2) showed that the spray-drying process was suitable for the production of amorphous microparticles from both polymers. The EN sd microparticles were larger (5.0–1.0 μm) than those of the EP sd (0.5–2.0 μm) but more collapsed due to the rapid solvent evaporation and with more aggregates, probably due to the carbohydrate nature of EN. The EP sd surface was smoother and more homogeneous than that of the EN sd. The atomization process improved the technological properties of all the polymers; in particular, the polymers became potentially more active in coating ingredients for oral administration. However, the reduction in the “particle size” increased the surface area, making the dissolution faster, a necessary step to ensure the correct in vivo absorption of any substance. Furthermore, a more homogeneous particle size distribution is better for predictions of pharmacokinetic behavior, as homogeneous particles are reported to be likely to interact easily with biological fluids and membranes.

The FTIR (Figure 3) and DSC (Figure 4) analyses of the SOE were carried out only with spray-dried extract due to the pure SOE’s soft and sticky nature, which prevented analysis. The SOE sd spectra showed characteristic peaks (-OH stretching at 3275.12 cm^−1^; C-H stretching at 2932.78 cm^−1^; aromatic fingerprinting from 1868.80 cm^−1^ to 1829.82 cm^−1^; C=O stretching vibrations from 1772.02 cm^−1^ to 1670.02 cm^−1^; C=C stretching vibrations from 1636.06 cm^−1^ to 1507.76 cm^−1^; 1488.99 cm^−1^–1472.63 cm^−1^ symmetric -CH_2_ bending vibrations; asymmetric -CH_3_ bending vibrations from 1456.95 cm^−1^ to 1418.70 cm^−1^; symmetric -CH3 bending vibrations from 1339.54 cm^−1^ to 1362.63 cm^−1^; C-C stretching from the phenyl group from 1300 cm^−1^ to 1450 cm^−1^; C-O stretching from 1244.07 cm^−1^ to 1075.17 cm^−1^) with signals ascribable to flavonoids. The region between 4000.0 cm^−1^ and 3000.0 cm^−1^ was identified as a flavonoid hydroxyl group signal; the regions at 1670.0 cm^−1^ and 1620.0 cm^−1^ were polyphenol saturation and conjugation bond signals, respectively; and 1650 cm^−1^ and 1600 cm^−1^ were identified as the characteristic aromatic signals of flavonoids [33,34]. Furthermore, SOE DSC showed a series of signals (from 230 °C to 270 °C) that were close together, probably ascribable to the blend of molecules belonging to the same class of polyphenols.

The FTIR spectra of the blank EN sd and EP sd microparticles (Figure 3) showed differences from the pure polymers. Large bands at 3397.75 cm^−1^ (EP sd) and 3295.46 cm^−1^ (EN sd) appeared in both FTIR spectra, ascribable to the -OH solvent group. In the EP sd, three other new bands were visible at 2614.31 cm^−1^and 2467.47 cm^−1^, probably due to the cleavage of the covalent bond between the carbonyl group and the close α-C atom, and at 1642.94 cm^−1^, where there was a new carbonyl group, different from those of the pure EP at 1722.50 cm^−1^. The signals at 1062.69 cm^−1^ and 1019.50 cm^−1^ were probably derived from the reduction in the 1182.38 cm^−1^ C-O stretching signal due to the scission of ether bonds with respect to carbonyl bonds. The bands at 1232.57 cm^−1^ and 1142.16 cm^−1^ were unchanged [35]. The pure EN showed characteristic signals: bands at 2156.27 cm^−1^ and 1473.00 cm^−1^ suggested the presence of an acrylonitrile group; bands at 2000.0 cm^−1^ and 1470.0 cm^−1^ were due to the stretching of a cyano group and symmetric bending of -CH2, respectively [36]. The EN sd FTIR spectra showed many differences. Three new bands were evident: at 1723.12 cm^−1^ and 1642.89 cm^−1^ due to carboxylic and amide groups derived from cyano-group hydrolysis, confirmed by the disappearance of the signal at 2000 cm^−1^ [37]; and at 2928.01 cm^−1^ due to C-H stretching and probable bond scission.

DSC analyses of both pure polymers (Figure 4) showed that spray-drying technology was capable of improving their characteristics. Two melting peaks (300.61 °C and 406.74 °C) and two glass transitions (58.50 °C and 87.84 °C) were visible in the pure EP thermogram. In contrast, the EP sd showed only one glass transition (75.81 °C) with dehydration and one melting peak (253.96 °C), both of which were shifted at low temperatures, undergoing a glassy–rubbery transition that may have improved the polymer dissolution and, consequently, the release of active content [38]. The EN DSC showed the gelatinization, retrogradation, and glass-transition phenomena of a starch-modified molecule [39]. As is well-known, a starch derivative consists of amorphous regions of amylose granules and amylopectin and crystalline regions due to the side chain of amylopectin. Therefore, the EN thermogram showed a semi-crystalline solid with initial glass transition at 52.0 °C and a visible dehydration phenomenon. Then, the melting of the amylopectin crystals between 274.0 °C and 286.0 °C was observed, with a degradation peak above 300.0 °C and the carbonization phenomenon above 500.0 °C [40].

#### 3.2.2. Characterization of EN- and EP-Based Microparticles Carrying SOE (EPSOE and ENSOE)

EN- and EP-based spray-dried microparticles carrying SOE were designed and developed using an aqueous solution of 1% EN (*w*/*v*) or an acidified aqueous solution (pH 1.0) of 1% EP (*w*/*v*) as feeds. SOE was added in a ratio of 1:1 *w*/*w* with respect to the polymers to obtain ENSOE and EPSOE microparticles. The IE value for all microparticles obtained was 100 ± 0.1% and all spray-dried yield values (65.0 ± 1.2 and 85.0 ± 2.3 for ENSOE and EPSOE, respectively) were similar to blanks. The lower yield for EN sd and its derived ENSOE microparticles than for EP sd and EPSOE was probably attributable to the starch nature of the EN. In fact, as for the SOE, the sugars could have adhered to the dry chamber, reducing the formation of micro-powders. However, the yield% of the EP-based microparticles could also be attributable to the smaller dimensions, as shown by SEM microphotographs (Figure 2). In particular, EPSOE microparticles were extremely small (0.5–3.0 μ). Despite all this, the process yield and encapsulation efficiency were considered good for all microparticles obtained. This was also due to the polymers used, which made it possible to partially avoid the separation phase that would have led to deposition of the extract in the spray chamber. This effect is generally observed for products containing sugars and flavonoids [35].

ENSOE and EPSOE also showed similar dimensions as the blank EN sd and EP sd, respectively, but with worse morphology. More aggregates were present, probably due to the presence of sugars, which could form hydrogen bonds on the surface of the microparticles, favoring inter-contact and causing aggregation. A few broken microparticles were found in EPSOE alone, while unencapsulated SOE was visible in ENSOE, confirming that EN was less capable than EP of coating SOE, as suggested by the FTIR and DSC analyses.

The ENSOE and EPSOE FTIR spectra (Figure 3) did not show extract–polymer interaction. Only -OH bands were enlarged due to -OH bond formation. ENSOE spectra also showed two characteristic signals of SOE (aromatic fingerprinting at 1868.76–1844.71 cm^−1^) due to the extract not being wholly coated by EN, as confirmed by the DSC analyses (Figure 4). In fact, ENSOE showed several SOE peaks at 260.0 °C and 270.0 °C, which were still visible at a lower intensity than for pure SOE. The EPSOE spectra did not show SOE or new peaks due to the complete encapsulation of the extract.

#### 3.2.3. Microparticle Release Profiles

The water dissolution/release profiles (Figure 5) showed an improvement in the SOE dissolution rate in the presence of both polymers. Only 23.0% of the SOE dissolved in 30 min in the unprocessed extract, while about 50.0% was released from EPSOE in a controlled manner and 80.0% from ENSOE over the same time (30 min).

It should be noted that ENSOE exhibited a profile with a burst effect (release of 60.0% in the first 5 min). The behavior of ENSOE may have been due to the EN, which, like other starch derivatives, is capable of swelling and eroding at the same time, acting as a super-disintegrant and reducing the hydrophobic interactions of SOE powders, thus improving the wettability of the extract and the in vitro dissolution/release [41].

### 3.3. In Vitro Biological Activity

Although *S. domestica* tree leaves are used as an antidiabetic remedy in traditional Mediterranean medicine [17], this effect has not been previously investigated. Thus, the hypoglycemic activity of SOE and its constituents was assayed in vitro against the α-glucosidase enzyme, a carbohydrate-hydrolyzing enzyme. Inhibition of α-glucosidase, which delays the intestinal absorption of glucose, limiting postprandial hyperglycemia, is used to manage and prevent diabetes mellitus type 2, a metabolic disorder of the glucose metabolism associated with several complications [19]. SOE exhibited a strong inhibitor effect on the α-glucosidase enzyme that was higher than that of acarbose, a well-known standard α-glucosidase inhibitor generally used as a positive control (Table 1). Moreover, the major isolated compounds inhibited the catalytic activity of α-glucosidase with different potencies (Figure 6). These findings suggested that the strong inhibitory activity of SOE against α-glucosidase could be related to its constituents’ synergistic or additive effects. It is evident that the SOE extract exerted a better hypoglycemic effect than that of its individual compounds.

Uncontrolled hyperglycemia produces free radicals and alteration of the antioxidant enzyme system, thus increasing tissue damage associated with diabetes mellitus pathology. Polyphenols can neutralize the intracellular production of free radicals, which may play a key role in ameliorating diabetes complications [19,42]. Thus, the capacity of SOE and its compounds to scavenge free radicals was evaluated in vitro against DPPH• radicals.

The results, summarized in Table 1, showed significant concentration-dependent free radical scavenging activity for SOE that may have been correlated with its high total polyphenolic content (66.4 ± 10 mg/g GAE of extract) and the structures of its isolated constituents. All tested compounds exhibited strong potency in scavenging free radicals, with EC_50_ values similar to or slightly lower than the positive control (Figure 6).

### 3.4. Accelerated Stability (ICH Guidelines) and Functional Stability

After one week at 40 °C, the actual active content (AAC), determined using HPLC and UV–Vis methods, showed a 30% decrease in the pure SOE extract (AAC ranging from 10.0% to 7.0%), while the AAC in the ENSOE and EPSOE microparticles remained relatively unaltered, showing that the polymers were able to protect the extract and could enhance SOE shelf life.

The reduction in the AAC in the SOE was probably due to the thermal degradation of some of the polyphenols contained in the extract. The FTIR spectra for the spray-dried SOE (SOE sd) showed fewer signals than the unprocessed SOE (Figure 7). In particular, signals of water content around 3900.0 cm^−1^ and from 1900.0 cm^−1^ to 2300.0 cm^−1^ and the signals from 1800.0 cm^−1^ to 1700.0 (aromatic rings and carbonyl stretching), from 1500.0 to 1600.0 cm^−1^ (C=C stretching), and from 1300.0 cm^−1^ to 1400.0 cm^−1^ (C-H bending) disappeared. The spray-drying technique and polymers used were able to protect the SOE from degradation; therefore, the FTIR spectra of the processed ENSOE and EPSOE were unaltered.

Similarly, the DSC of spray-dried SOE indicated the disappearance of signals attributable to water evaporation and the signals from 230.0 °C to 260.0 °C. Similar behavior was shown in the ENSOE thermogram. However, the DSC of the processed EPSOE was superimposable with that of the unprocessed EPSOE (Figure 8).

To verify the capacity of the chosen spray-dried conditions to preserve the functional activity of SOE, the α-glucosidase inhibition enzyme and the free radical scavenging activity (DPPH• test) were investigated after the microencapsulation process (t0) and under harsh storage conditions (t7 days). EPSOE and ENSOE inhibited the carbohydrate-hydrolyzing enzyme and scavenged the DPPH• radical with a potency higher than that of the raw material (Table 1), probably due to the protection of the SOE by the polymers. Indeed, no reduction in activity for the SOE microsystems during the accelerated storage condition (Table 1) was observed, while the SOE raw material lost its functional activity and the AAC decreased (Table 1). As reported elsewhere [43], the selected carriers associated with the spray-dried process are able to preserve the functional activity of polyphenol-rich extracts.

## 4. Conclusions

EN and EP are two innovative “GRAS” polymers. They are capable of vehiculating and protecting botanicals from oxidation and degradation, improving their technological and biological characteristics. These properties were applied to the microencapsulation of SOE, an aqueous extract of *S. domestica* L. leaves rich in polyphenols that is not cytotoxic, as established by MTT assay.

Unfortunately, SOE was difficult to handle, showed low solubility, and exhibited degradation phenomena.

The data collected and the information obtained with the DSC FTIR spectra and SEM micrographs showed that the EN and EP were, with the employment of spray-drying technology, able to encapsulate SOE with appreciable production yields and encapsulation efficiency for all designed microsystems. The actual active content (AAC) was very close to the theoretical composition, and total coating and microencapsulation of the extract was demonstrated.

In the presence of the EP, a significant improvement in the in vitro water dissolution, with constant release of the extract over time (30 min), was observed.

In contrast, EN, as a starch derivative that swells in water, may act as a super-disintegrant, improving the SOE wettability and dissolution rate with an initial burst effect (60.0% release in 5 min).

Both polymers have properties that are able to improve end-product shelf life, reducing SOE stickiness, providing better handling, and improving its antioxidant activity and α-glucosidase inhibition. These results show that SOE is an excellent candidate for the formulation of dietary supplements designed for type 2 mellitus diabetes patients.

## Figures and Tables

**Figure 1 pharmaceutics-15-00295-f001:**
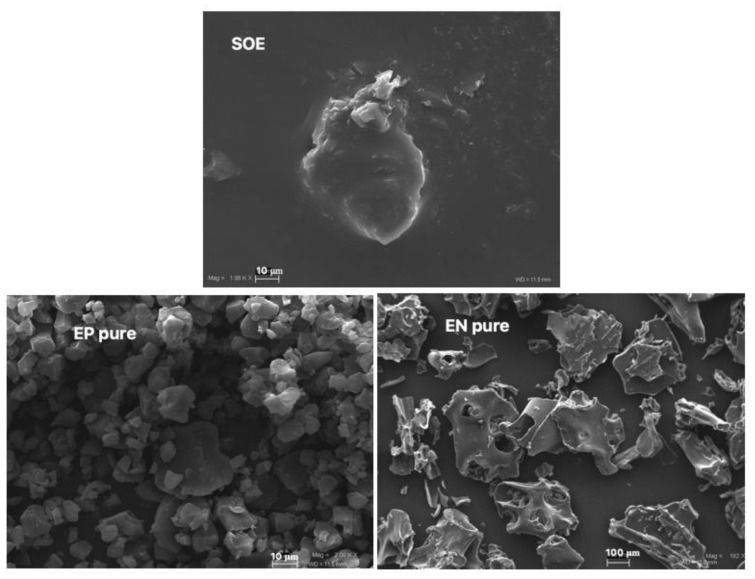
Unprocessed S. domestica leaves extract (SOE) and raw polymers (pure EN, pure EP) at different magnifications (mag): SOE, 2.49 KX; pure EP, 1.82 KX; pure EN, 182 X. Frame average, N = 1.

**Figure 2 pharmaceutics-15-00295-f002:**
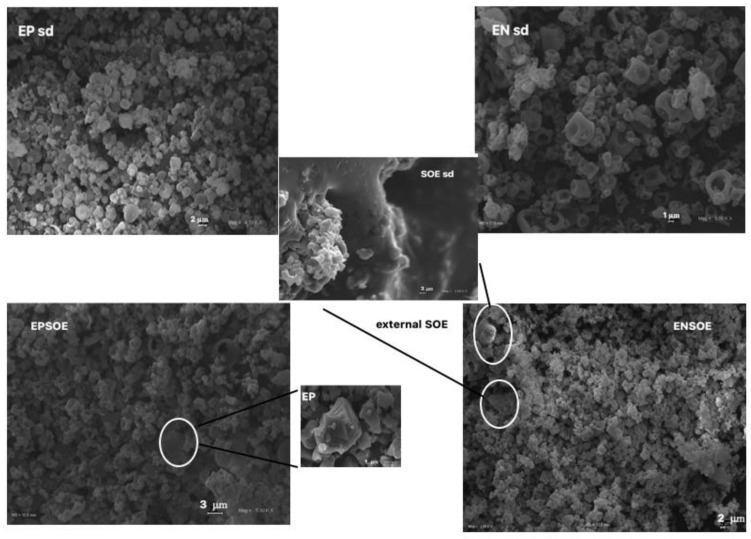
SEM microphotographs of blank (EP sd and EN sd) and loaded SOE microparticles (EPSOE and ENSOE) compared to processed SOE (SOE sd) and pure EP (EP) at different magnifications (mag): SOE sd, 2.49 KX; EP, EPSOE, and EN sd, 5.0 KX; EP sd, 4.18 KX; ENSOE, 2 KX. Frame average, N = 1.

**Figure 3 pharmaceutics-15-00295-f003:**
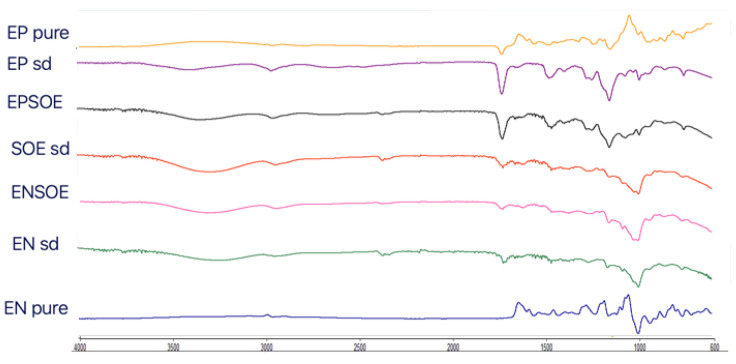
FTIR spectra of raw materials (pure EP, pure EN, SOE sd) and blank (EP sd, EN sd) and loaded (EPSOE, ENSOE) microparticles.

**Figure 4 pharmaceutics-15-00295-f004:**
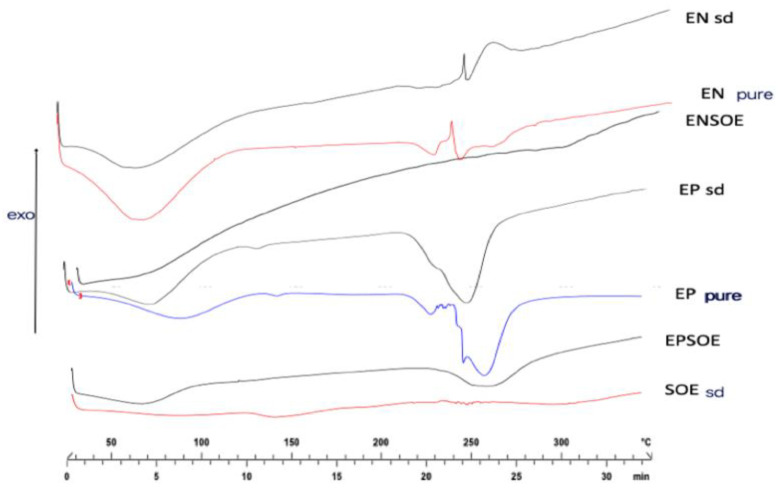
DSC thermograms of polymer raw materials (pure EP, pure EN) and blank (EP sd, EN sd) and loaded (EPSOE, ENSOE) microparticles.

**Figure 5 pharmaceutics-15-00295-f005:**
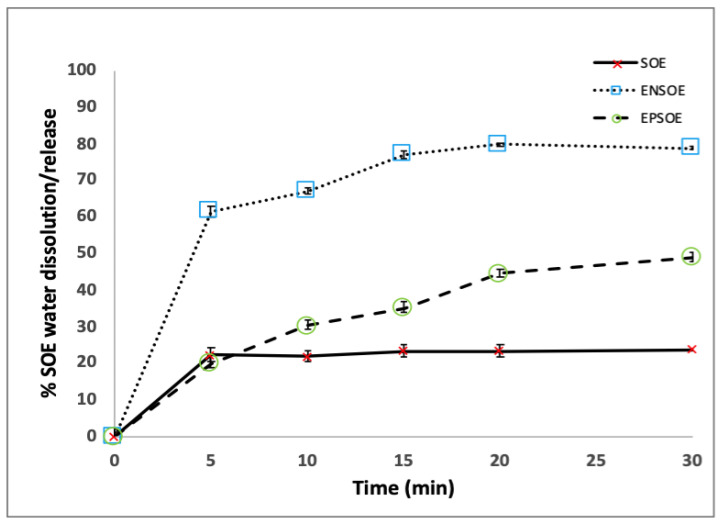
Dissolution/release profile of SOE, EPSOE, and ENSOE in water.

**Figure 6 pharmaceutics-15-00295-f006:**
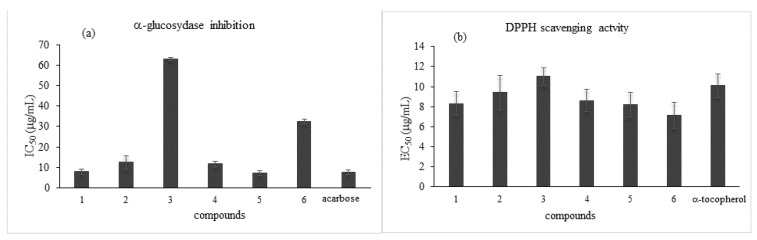
α-Glucosidase inhibition enzyme (**a**) and DPPH scavenging activity (**b**) in compounds isolated from SOE extract.

**Figure 7 pharmaceutics-15-00295-f007:**
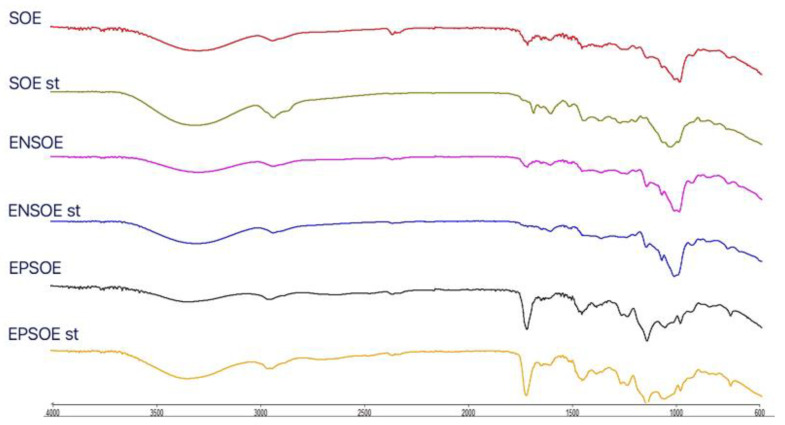
FTIR spectra of unprocessed (SOE sd, ENSOE, EPSOE) and processed (SOE st, ENSOE st, EPSOE st) materials.

**Figure 8 pharmaceutics-15-00295-f008:**
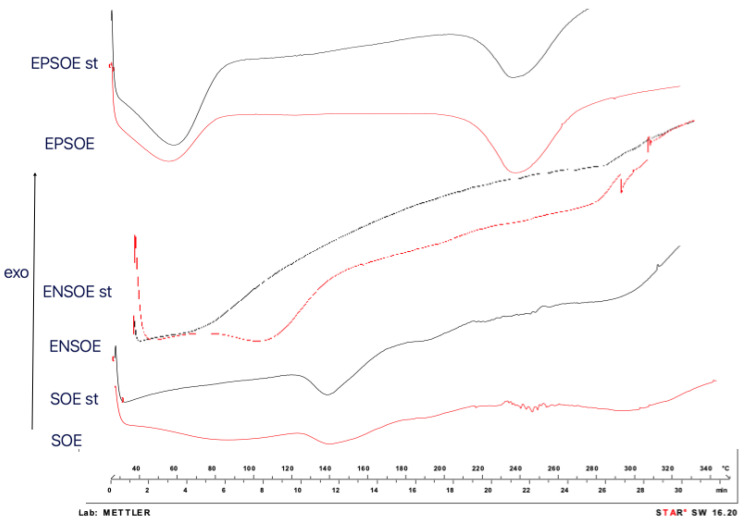
DSC thermograms of unprocessed (SOE sd, ENSOE, EPSOE) and processed (SOE st, ENSOE st, EPSOE st) materials.

**Table 1 pharmaceutics-15-00295-t001:** HPLC actual active content (AAC), free-radical scavenging activity (DPPH assay), and α-glucosidase inhibition for unprocessed SOE extract and EPSOE and ENSOE microsystems.

	AAC% ^a,b^				DPPH Assay EC_50_ ^a,c^		α-Glucosidase IC_50_ ^a,c^	
	T_0_		T_7days_		T_0_	T_7day_	T_0_	T_7day_
Samples	Qcn	Rt	Qcn	Rt				
SOE	2.4 ± 0.1	2.9 ± 0.2	1.7 ± 0.4	2.2 ± 0.2	24.1 ± 1.9	30.1 ± 1.4	4.3 ± 0.8	5.6 ± 1.4
EPSOE	2.7 ± 0.2	2.9 ± 0.3	2.7 ± 0.2	2.8 ± 0.4	24.5 ± 1.6	25.4 ± 0.8	4.8 ± 1.1	5.1 ± 0.9
ENSOE	2.4 ± 0.6	2.8 ± 0.2	2.3 ± 0.3	2.7 ± 0.4	23.3 ± 0.9	24.5 ± 1.1	4.2 ± 1.0	4.4 ± 0.5
α-Tocopherol ^d^					10.1 ± 1.3			
Acarbose ^d^							7.5 ± 1.7	

All results are expressed as means ± standard deviation (SD) from three experiments performed in triplicate. ^a^ Actual active content (AAC) was determined as the concentration of quercitrin (Qcn) and rutin (Rt) determined with HPLC and expressed as the percentage in 100 mg of powder; ^b^ EC_50_ ± SD, expressed as a μg unit of SOE or ENSOE and EPSOE/mL; ^c^ IC_50_ ± SD, expressed as a μg unit of SOE or ENSOE and EPSOE/mL; ^d^ α-tocopherol and acarbose were used as positive controls for the DPPH and α-glucosidase assays, respectively.

## Data Availability

Not applicable.

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
