# Peer review of "Eudraguard® Natural and Protect: New “Food Grade” Matrices for the Delivery of an Extract from Sorbus domestica L. Leaves Active on the α-Glucosidase Enzyme"

_pharmaceutics, 2023, doi:10.3390/pharmaceutics15010295_

Round 1

Reviewer 1 Report

In this article, two polymers have been evaluated to functionalise an extract from Sorbus domestica. Different characterisation techniques have been performed including HPLC, SEM, FTIR and DSC. The utilisation of this extract to treat diabetes and testing the biological activity are an interesting alternative to conventional treatments, however, I have concerns about the safety, as it has not been tested and plant extracts usually contain other compounds which can compromise the efficacy and safety of the desired compound. Apart from that, the article contains grammar errors that should be carefully checked and some figures need improvement. Please check the following comments, that hopefully will help with that:

·         The links from Evonik should appear in the reference list, not through the text.

·         Line 78: ‘in September 2020’ instead of ‘in the September 2020’.

·         Line 102: ‘develop’ instead of ‘developed’.

·         Section 2.2.1.1: the authors should not be reporting results in this section.

In this section, I’m not sure if reporting the standard calibration curve is needed.

Validation, precision and linearity methods not needed.

Authors should improve the description of the morphology studies. They only mentioned they did SEM and the parameters, please rewrite.

·         Line 117: please report what TLC means.

·         Line 130: ‘was analysed’ instead of ‘were analysed’.

·         Line 135: TPC is total phenolic content? Please clarify.

·         Line 170: p/V? Do the authors mean w/v?

·         Section 2.2.3: particle size was measured based on SEM images. Why didn’t the authors measure particle size with appropriate particle size analysis techniques?

·         Line 219: ‘were evaluated’ instead of ‘was evaluated’.

·         Line 234: ‘HPLC, FTIR and DSC’ instead of ‘HPLC, FTIR e DSC’.

·         Line 252: is MS mass spectrometry? Please clarify.

·         Table 1: be careful when reporting. There is no ‘c’ in the table legend. There is a spiral instead of an α in α-tocopherol. Also, it’s not clear what Qcn and Rt mean and the table overall is difficult to understand.

·         Line 294: 0.24.0?

·         Line 326: ‘were identified’ instead of ‘were identify’.

·         Figure 3 looks messy. It’d be easier to see an understand by combining all spectrograms in one graph and deleting the peak labels.

·         Figure 4: please use a larger font size for the X axis.

·         Figure 5: statistics is missing.

·         Figure 7 would also look better if the peak labels were removed.

Author Response

R. In this article, two polymers have been evaluated to functionalise an extract from Sorbus domestica. Different characterisation techniques have been performed including HPLC, SEM, FTIR and DSC. The utilisation of this extract to treat diabetes and testing the biological activity are an interesting alternative to conventional treatments, however, I have concerns about the safety, as it has not been tested and plant extracts usually contain other compounds which can compromise the efficacy and safety of the desired compound. Apart from that, the article contains grammar errors that should be carefully checked and some figures need improvement. 

A.  Dear Reviewer 1,

Thank you very much for your encouragement and precious suggestions.

We revised and improved the manuscript as suggested. An extensive English language was also revised.

R. Please check the following comments, that hopefully will help with that:

The links from Evonik should appear in the reference list, not through the text.

The links to Evonik have been moved to the reference list

R. Line 78: ‘in September 2020’ instead of ‘in the September 2020’.

A. This point has been corrected

R. Line 102: ‘develop’ instead of ‘developed’.

A. The grammar error has been corrected

R. Section 2.2.1.1: the authors should not be reporting results in this section.

In this section, I’m not sure if reporting the standard calibration curve is needed.

Validation, precision and linearity methods not needed.

Authors should improve the description of the morphology studies. They only mentioned they did SEM and the parameters, please rewrite.

R: Section 2.2.1 has been modified. Results have been moved to the appropriate section; Validation, precision, and linearity methods have been deleted and SEM studies have been improved.

R. Line 117: please report what TLC means.

A. Thank you for the suggestion; TLC (Thin Layer Chromatography) has been added

R. Line 130: ‘was analysed’ instead of ‘were analysed’.

A. The grammar error has been corrected

R. Line 135: TPC is total phenolic content? Please clarify.

A. Thank you for the suggestion, TPC (total phenolic content) has been added

R. Line 170: p/V? Do the authors mean w/v?

A. Sorry for the mistake; this really is w/v

R. Section 2.2.3: particle size was measured based on SEM images. Why didn’t the authors measure particle size with appropriate particle size analysis techniques?

A. The particle size was also measured by laser scattering technology (Beckman Coulter LS 13320). EN can be analyzed in isopropanol (see figure below); on the contrary, EP rapidly swells both from H2O and EtOH (the only solvents compliant with Coulter cells preventing proceeding with the analysis (see attached file).

R. Line 219: ‘were evaluated’ instead of ‘was evaluated’.

A: We apologize for the error; it has been corrected

R. Line 234: ‘HPLC, FTIR and DSC’ instead of ‘HPLC, FTIR e DSC’.

A. The mistake at line 234 has been corrected

R. Line 252: is MS mass spectrometry? Please clarify.

A. The acronym “MS” has been modified in EIMS and clarified in the text.

R. Table 1: be careful when reporting. There is no ‘c’ in the table legend. There is a spiral instead of an α in α-tocopherol. Also, it’s not clear what Qcn and Rt mean and the table overall is difficult to understand.

A. Table 1 has been modified, and symbols and acronyms have been corrected.

R. Line 294: 0.24.0?

A. Thank you, the value has been corrected (24.0).

R. Line 326: ‘were identified’ instead of ‘were identify’.

A. Thank you, the grammar error has been corrected.

R. Figure 3 looks messy. It’d be easier to see an understand by combining all spectrograms in one graph and deleting the peak labels.

A. Thank you for the suggestion, the peak labels have been removed, and spectrograms combined.

R. Figure 4: please use a larger font size for the X axis.

A. Thank you for the suggestion; a larger font size has been used

R. Figure 5: statistics is missing.

A. The statistic analysis was inserted in the first manuscript. The error bars are probably not noticeable due to the low value. Therefore, the graph has been modified, colored and enlarged

R. Figure 7 would also look better if the peak labels were removed.

A.Thank you for the suggestion; the peak labels have been removed.

Reviewer 2 Report

- Undoubtedly, this is a job of great importance.

- Aspects of form must be taken care of, for example the name of the plants and the Latinisms will have to be put in italic font.

- Take care of aspects of writing in the English language, for example in line 115 they write chromatographed, as if it were a verb.

- I consider that each of the tests carried out were those necessary for a work of this type.

- It would be worth including in the discussion aspects of the non-toxicity of the EUDRAGUARD® product.

Author Response

R. Undoubtedly, this is a job of great importance.

A. Dear Reviewer 2,

Thank you for your positive comments and suggestions. The manuscript has been revised and improved as suggested.

R. Aspects of form must be taken care of, for example the name of the plants and the Latinisms will have to be put in italic font.

A. Thank you for the suggestion. All the Latinisms have been put in italic font

R. Take care of aspects of writing in the English language, for example in line 115 they write chromatographed, as if it were a verb.

A: Thank you, a careful revision of the English language has been made

R. I consider that each of the tests carried out were those necessary for a work of this type.

A. Thank you for your comment.

R. It would be worth including in the discussion aspects of the non-toxicity of the EUDRAGUARD®product.

A. The non-toxicity of Eudraguard® according to EUFAPS and FDA, was introduced in the discussion, as suggested.

Reviewer 3 Report

The present work reports the use of two matrices, Eudraguard® Natural and Protect, to functionalize an extract obtained from Sorbus domestica leaves. Several approaches were applied to physico-chemical characterization, and both antioxidant and alpha-glucosidase inhibitory activities were assessed. In my opinion, the manuscript fits with the scope of Pharmaceutics, although requires mandatory corrections.

About title, I recommend to remove the sentence "...Active on Type-2 Mellitus Diabetes". The idea of the obtained products were assayed in an experimental model of diabetes comes at a glance. Furthermore, any background regarding diabetes and the role of previous reports on Sorbus domestica were cited. 

Thus, about Introduction, all the text are merged in only one paragraph, an then it is not good to follow. Please, split the text in some paragraphs as needed in order to summarize the background, hypotesis and aims. 

Lines 87-88: please add the meaning of "CD3OD" and "TMS" in parenthesis.

Line 100: Subtopic 2.2 is repeated. Please correct numeration.

Line 215: GraphPad Prism must be accompanied by city and country.

Line 215: EC50 must be "EC50". Please revise throughout the manuscript.

Line 220: Should it be "mixture containing 10 µL"...

Lines 87-88: please add the meaning of "CD3OD" and "TMS" in parenthesis.

Line 100: Subtopic 2.2 is repeated. Please correct numeration.

Line 215: GraphPad Prism must be accompanied by city and country.

Line 229: IC50 must be "IC50". Please revise throughout the manuscript.

Author Response

R. The present work reports the use of two matrices, Eudraguard® Natural and Protect, to functionalize an extract obtained from Sorbus domestica leaves. Several approaches were applied to physico-chemical characterization, and both antioxidant and alpha-glucosidase inhibitory activities were assessed. In my opinion, the manuscript fits with the scope of Pharmaceutics, although requires mandatory corrections.

A. Dear Reviewer 3,

Thank you for your revision. We accepted all of your suggestions and, in accordance, improved the manuscript.

R. About title, I recommend to remove the sentence "...Active on Type-2 Mellitus Diabetes". The idea of the obtained products were assayed in an experimental model of diabetes comes at a glance. Furthermore, any background regarding diabetes and the role of previous reports on Sorbus domestica were cited.

A. The authors aimed to focus on antidiabetic activity since the only data in the literature concerned the fruits and not the leaves. According to your suggestion, the title has been modified as follows

“Eudraguard® Natural and Protect: New "Food Grade" Matrices for the delivery of an Extract from Sorbus Domestica L. Leaves active on the a-glucosidase enzyme".

R. Thus, about Introduction, all the text is merged in only one paragraph, an then it is not good to follow. Please, split the text in some paragraphs as needed in order to summarize the background, hypotesis and aims.

R. Following the guidelines of Pharmaceutics which do not require the division of the introduction into paragraphs, we applied the guidelines only for the abstract. In any case, based on your suggestion, the introduction has been revised and reorganized into paragraphs.

R. Lines 87-88: please add the meaning of "CD3OD" and "TMS" in parenthesis.

A. Thank you, means of the acronyms "CD3OD" and "TMS" have been added

R. Line 100: Subtopic 2.2 is repeated. Please correct numeration.

A. Thank you, the numeration has been corrected

R. Line 215: GraphPad Prism must be accompanied by city and country.

A. Thank you, City and Country were added

R: Line 215: EC50 must be "EC50". Please revise throughout the manuscript.

A. Thank you, EC50 has been changed in "EC50" in all manuscript

R: Line 220: Should it be "mixture containing 10 μL"...????????

A. The sentence has been adjusted.

R. Line 229: IC50 must be "IC50". Please revise throughout the manuscript.

A. Thank you, IC50 has been changed in "IC50" in all manuscript

Round 2

Reviewer 3 Report

Suggested corrections were performed.